# A Resource-Friendly Certificateless Proxy Signcryption Scheme for Drones in Networks beyond 5G

**Muhammad Asghar Khan** [1,*], **Hosam Alhakami** [2], **Insaf Ullah** [1], **Wajdi Alhakami** [3], **Syed Agha Hassnain Mohsan** [4], **Usman Tariq** [5] **and Nisreen Innab** [6]

1   Department of Electrical Engineering, Hamdard Institute of Engineering and Technology, Hamdard University, Islamabad 44000, Pakistan; insaf.ullah@hamdard.edu.pk
2   Department of Computer Science, College of Computer and Information Systems, Umm Al-Qura University, Makkah 24231, Saudi Arabia; hhhakam@uqu.edu.sa
3   Department of Information Technology, College of Computers and Information Technology, Taif University, Taif 21431, Saudi Arabia; whakami@tu.edu.sa
4   Optical Communications Laboratory, Ocean College, Zhejiang University, Zheda Road 1, Zhoushan 316021, China; hassnainagha@zju.edu.cn
5   Department of Management Information Systems, College of Business Administration, Prince Sattam Bin Abdulaziz University, Al-Kharj 16278, Saudi Arabia; u.tariq@psau.edu.sa
6   Department of Computer Science and Information Systems, College of Applied Sciences, AlMaarefa University, P.O. Box 71666, Riyadh 11597, Saudi Arabia; ninnab@mcst.edu.sa
*   Correspondence: m.asghar@hamdard.edu.pk

**Abstract:** Security and privacy issues were long a subject of concern with drones from the past few years. This is due to the lack of security and privacy considerations in the design of the drone, which includes unsecured wireless channels and insufficient computing capability to perform complex cryptographic algorithms. Owing to the extensive real-time applications of drones and the ubiquitous wireless connection of beyond 5G (B5G) networks, efficient security measures are required to prevent unauthorized access to sensitive data. In this article, we proposed a resource-friendly proxy signcryption scheme in certificateless settings. The proposed scheme was based on elliptic curve cryptography (ECC), which has a reduced key size, i.e., 160 bits, and is, therefore, suitable for drones. Using the random oracle model (ROM), the security analysis of the proposed scheme was performed and shown to be secure against well-known attacks. The performance analysis of the proposed scheme was also compared to relevant existing schemes in terms of computation and communication costs. The findings validate the practicability of the proposed scheme.

**Keywords:** drones; security; proxy signcryption; hyperelliptic curve cryptography; random oracle model; 5G

## 1. Introduction

Drones were considered for several applications and case studies because of their flexible flight capabilities, which include flying at low altitudes, at high elevation angles, and over urban, suburban, and rural areas [1]. The typical drone is outfitted with all of the electronic components required to carry out its mission in an efficient manner. These components include a communication module for transmitting data to the ground station (GS), sensors for gathering data, memory for storing the information collected by the sensors, as well as computational and power resources to process information and maintain flight for a predetermined amount of time, respectively [2–5]. In addition, recent advancements in fifth-generation (5G) wireless communications made possible the concept of cellular networks beyond 5G (B5G), which may fully unlock the promise of autonomous services and provide wide coverage for drones. Drones outfitted with AI systems in flight would be possible with the faster data transfer rates made possible by a B5G network. The most important development in B5G is satellite integration, which allows drones to deliver centimeter-

level precise positioning, global coverage, and heterogeneous QoS provisioning [6,7]. If legal provisions permit drone integration with B5G and autonomous flying, the sky will be filled with drones performing activities such as mail and package delivery, traffic monitoring, event filming, surveillance, search and rescue, and marine monitoring [8].

Drones are typically not designed with security and privacy concerns in mind, leaving them vulnerable to both cyber and physical attacks [9–11]. Intruders who wish to compromise the security and privacy of a drone have a variety of options. They could, for instance, send out numerous reservation requests, eavesdrop on control communications, and/or forge data exchange [12]. Due to unreliable connections and insufficient security protocols, anyone with the proper transmitter can attach to a drone and embed commands into an ongoing session, making them readily interceptable [13]. If drones fly over a hostile environment, they could become an enticing target for physical attacks. This is another security and privacy concern. In these situations, an intruder can deceive captured drones to gain access to their internal data via standard interfaces or terminals.

Many drone systems depend on the GNSS (global navigation satellite system) for precise location, navigation, and timing for safe and efficient operation. GLONASS, Galileo, BeiDou, and NavIC are also used in drones, although global positioning system (GPS) is the most common. All GNSS systems are subject to cyber-physical attacks [14,15]. For example, GPS spoofing [16] is another significant security threat that occurs when an adversary manipulates the drone's GPS signals. In this attack, an adversary transmits fake GPS signals to an intended drone at a slightly higher frequency than the real GPS signals, so that the drone believes it is located elsewhere. In B5G networks, however, drones can be linked to new wireless technologies such as visible light communications and quantum communications, which could introduce new security threats [1]. The best GNSS system for a drone application depends on the use case, precision, and dependability needed, and system risks and vulnerabilities. Drone operators should be aware of GNSS attack threats and take precautions such as employing backup navigation systems or secure communication methods. Additional security mechanisms and countermeasures will be necessary to combat such security hazards.

The deployment of non-terrestrial infrastructures as part of the B5G network, also known as the integrated space and terrestrial drone networks, is regarded as a topic of the long term with the aim of improving coverage rates [17]. Figure 1 depicts a typical drone architecture for B5G networks, which may include drones, a command center (CC), ground control stations (GCS), and satellites. The stations that can command drones are the CCs, GCSs, and satellites. When a CC intends to issue direct mission commands to drones, digital signcryption ensures the commands' authenticity, integrity, and confidentiality. However, a drone occasionally performs remote tasks beyond the CC's range, preventing the CC from communicating directly with the drones. In this scenario, the CC designates a GCS in the drone's proximity as an agent, and the GCS transmits commands directly to the drone. A proxy signcryption scheme can be used to achieve confidentiality and authentication of the transmitted commands and ensure the drone executes commands in a timely manner.

Using either the public key infrastructure (PKI), a certificateless cryptosystem (CLC), or an identity-based cryptosystem (IBC), the proxy signcryption scheme can be developed. PKI's primary shortcoming is that its standard application cannot be used with drones. Certificate management overhead, such as certificate storage, distribution, and revocation, is the crucial factor that renders them unsuitable for drone systems [18]. IBC [19] is implemented to alleviate the burden on conventional PKI, which uses a publicly recognized string as a public key to reduce the cost of PKI certificate renewal. Being identity-based, the IBC appears to be more vulnerable to external infiltration (key escrow problem). CLC [20] was developed to address these problems. Key generation center (KGC) generates and distributes partial private keys to network participants. The user will then generate his or her own private and public keys by combining a private key fragment with some arbitrarily generated integers. Therefore, certificateless proxy signcryption scheme is the

optimal solution for drones' system requirements. As a result, we strengthened the proxy signcryption scheme with the following new features, which represent our most significant contributions:

- We propose a resource-friendly certificateless proxy signcryption scheme for drones in B5G networks. The proposed scheme is based on the elliptic curve cryptography (ECC) algorithm and enjoys some of its favorable features, such as no key escrow and no secure channel.
- The proposed scheme has a clear distribution of roles: the control center acts as the original signer, the network provider serves as the key generation center (KGC), the ground control station acts as a proxy, and the drones perform the task of unsigncryption.
- The proposed protocol guarantees anonymity for both senders and receivers by employing a mechanism wherein participants $(PP_i)$, where $i = (CC, GCS, drone)$, send their identities in an encrypted form while requesting a partial private key.
- The proposed scheme is capable of withstanding a wide variety of commonly known attacks under ROM. Additionally, it was found that this scheme is efficient in terms of both computation and communication costs when compared to other existing schemes.

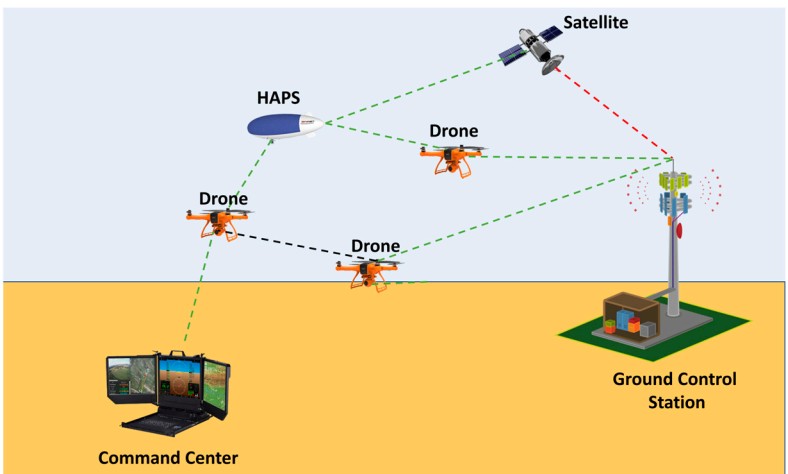

**Figure 1.** A typical drone architecture.

The remainder of this article is structured as follows. The literature review is covered in Section 2. Section 3 explains the network model and construction of the proposed scheme. The security analysis is discussed in Section 4. Section 5 discusses performance analysis. The conclusion of the proposed work is discussed in Section 6.

## 2. Literature Review

In 1996, Mambo et al. [21] were the first to introduce the concept of a proxy signature scheme. The proxy signature scheme was founded on the concept that the original signer delegates signing authority to the proxy signer, who then issues a legitimate signature on behalf of the original signer. Proxy signcryption is a combination of the proxy signature concept and the signcryption algorithm. In this technique, an original signcrypter is responsible for delegating the rights of his signcryption to a proxy signcrypter, who then signcrypts the message on behalf of the original signcrypter. The receiver is responsible for decrypting the signcrypted message in order to retrieve its contents and determining whether or not the signcrypted message is legitimate after receiving it. Gamage et al. [22] first presented the idea of proxy signcryption. Using an effective proxy signcryption scheme can provide a strong protection against attacks.

Yu and Wang [23] designed a certificateless proxy signcryption (CLPSC) scheme from CMGs. In the random oracle model (ROM), the authors demonstrated that their

scheme [23] had indistinguishability under adaptive chosen-ciphertext attacks (IND-CCA2 security) and existential unforgeability under adaptive chosen-message attacks (UF-CMA security). Abdelfatah [24] introduced a novel proxy signcryption scheme that utilized the ECC algorithm. However, the author of this novel proxy signcryption scheme did not provide a formal security analysis and lacked an anti-replay attack security property. Yanfeng et al. [25] developed a certificateless proxy identity-based signcryption scheme without bilinear pairings by combining the certificateless signcryption and proxy signcryption. This scheme had a strong foundation in the elliptic curve discrete logarithm problem, which provides a high level of security. This scheme was efficient and secure as it only required some operations on elliptic curves, without the need for pairing computation.

Bhatia and Verma [26] performed a cryptanalysis on the scheme devised by Yanfeng et al. [25] and demonstrated that it was vulnerable to a forgery attack. In addition, they offered a protected proxy signcryption scheme. In contrast, the scheme did not guarantee security services such as forward security and anti-replay attack. The public key replacement attack was also susceptible to exploiting this vulnerability. However, Li et al. [27] first demonstrated that Bhatia and Verma's [26] scheme was vulnerable to a public key replacement attack, and then presented a new certificateless proxy signcryption scheme. However, the scheme was hindered by the requirement of a secure link for the distribution of the participants' partial private keys. It also lacked forward security and protection against replay attacks. Finally, Y. Qu and J. Zeng [28] proposed a certificateless proxy signcryption for the DRONE network under standard model. This scheme accomplished public verifiability and was EUF-CMA-CLPSC secure and IND-CCA-CLPSC-secure against adversaries of types I and II. However, the proposed scheme incurred substantial computation and communication costs.

This work proposes a certificateless proxy signcryption scheme that is specifically designed for drones, offering a promising solution to address the shortcomings of previous schemes. The proposed scheme has a great advantage in that the partial private key can be distributed through an open network without any risk of being disclosed to an outsider adversary. Additionally, the proposed scheme is highly efficient in terms of computation and communication costs.

## 3. Preliminaries

This section explains the random oracle model (ROM), adversarial or threat model, syntax, and network model of the proposed certificateless proxy signcryption scheme.

### 3.1. Random Oracle Model

In 1993, Bellare and Rogaway created the random oracle model (ROM). By considering hash functions as random oracles, this model makes it simple to verify the security of cryptographic algorithms that use hash functions. In this paradigm, any input will result in an output of a predetermined length. If the input was requested previously, the oracle returns the same value as it did previously. If the input is not one that the oracle previously received, the oracle returns a randomly chosen output. You can substitute a hash function with an accessible random function (the "random oracle"). Therefore, an adversary must consult the random number generator to determine what the hash function will do.

### 3.2. Adversarial or Threat Model

This section will outline potential security vulnerabilities that could compromise the confidentiality of the security parameter utilized in the generation of ciphertext and signatures. Two types of adversaries, namely Type 1 Type 1 ($NP_{A_1}$) and Type 2 ($NP_{A_2}$), are defined. The first type of attacker, denoted as Type 1 ($NP_{A_1}$), is an external threat actor who aims to compromise the confidentiality of the proposed scheme and engage in signature forgery. It should be noted that the entity denoted as ($NP_{A_1}$) lacks the capability to access the private keys of the user, yet possesses the ability to replace the public key of said user.

Type 2 ($NP_{A_2}$) is the insider attacker (malicious $NP_{KC}$) who desires to violate confidentiality and falsify the signature of the proposed scheme. It should be noted that the entity denoted as $NP_{A_2}$ possesses the capability to access the private key of $NP_{KC}$, yet lacks the ability to substitute the public key of the user. The primary objective for both adversaries is to reveal the parameters utilized in the creation of the secret key and ciphertext. The subsequent objective entails the construction or retrieval of parameters utilized in the computation of a signature, followed by the generation of a forge signature.

### 3.3. Syntax of Certificateless Proxy Signcryption

The syntax of the proposed scheme contains the following steps.

*Setup:* Here, the network provider ($NP_{KC}$) assumes the role of KGC; when it receives the security parameter $k_{KC}$, $NP_{KC}$ generates his private key ($\Phi_{KC}$), his public key ($\delta_{KC}$), and public parameters set $PAR_{KC}$.

*Partial Key Generation (PCGU):* The participant ($PP_i$) desires a partial private key ($P_i$) from $NP_{KC}$, first, it sends $\left(EID_{PP_i}, U_{PP_i}\right)$ through insecure network to $NP_{KC}$. Alternatively, when $NP_{KC}$ receives $\left(EID_{PP_i}, U_{PP_i}\right)$, it generates and sends $EP_i$ as an encrypted partial private and public key to $PP_i$ over an insecure network.

*Public and Private Key Generation (PBCGU):* When $PP_i$ receives $EP_i$, it sets ($U_{PP_i}, X_{KC}$) to his public key and sets ($P_i, \beta_{PP_i}$) to his private key.

*Delegation Generation (DG):* This phase is run by the CC and when it receives $PAR_{KC}$, ($ID_{CC}, ID_{CG}, P_{CC}, \beta_{CC}, U_{CG}, X_{CG}$), where $ID_{CC}$ and $ID_{GCC}$ are the identities of CC and GCS, respectively, ($P_{CC}, \beta_{CC}$) is CC's private key pair, and ($U_{CG}, X_{CG}$) is GCS's public key pair. After that, CC generates and sends the triple $\left(m_w, S_{m_w}, O_{CC}\right)$ as a delegation to the GCS through an open network.

*Delegation Verification (DV):* When $\left(m_w, S_{m_w}, O_{CC}\right)$ is received by the GCS, then it can perform verification procedures, to check whether the signature is valid or not.

*CL-Proxy Signcryption Generation (CL-PSG):* This phase is executed by the GCS, which generates and sends the triple ($C_{GCS}, S_{GCS}, Q_{GCS}$) as a proxy signcryption to *drone* via an open network.

*CL-Proxy Un-Signcryption (CL-PU-S):* When ($C_{GCS}, S_{GCS}, Q_{GCS}$) is received by the drone, it performs the verifications steps, to check whether the signature is valid or not, if the signature is valid, it performs decryption process to recover plaintext from ciphertext.

### 3.4. Network Model

The command center (CC), ground control stations (GCS), high altitude platform system (HAPS), drones, and satellites comprise the network architecture for the proposed scheme. Each drone is equipped with a variety of useful components, including cameras, a global positioning system (GPS), an inertial measurement unit (IMU), and sensors, which can be utilized in a variety of application scenarios. In contrast to terrestrial communication systems, satellite services depend on geostationary satellites to transmit and receive signals in out-of-range regions. Additionally, HAPS provides greater coverage/relay and interacts with satellites, enabling more reliable drone communication networks, particularly when satellite communications are disrupted by inclement weather. HAPS may utilize B5G, and the drones require no additional equipment.

The networks depicted in Figure 1 comprise of several stations, namely the CCs, GCSs, HAPs, and satellites, which possess the capability to command drones. Digital signcryption ensures the authenticity, confidentiality, and integrity of direct mission commands issued by a CC to drones. Occasionally, a drone conducts remote duties beyond the CC's range, preventing direct communication between the CC and the drones. In this scenario, the CC identifies a nearby GCS as an agent, and the GCS transmits commands directly to the drone. The proposed scheme operates under the assumption that HAPs function as the Key Generation Center (KGC).

## 4. Construction of the Proposed Scheme

The proposed scheme is comprised of seven algorithms: Setup, Partial Key Generation (PCGU), Public and Private Key Generation (PBCGU), Delegation Generation (DG), Delegation Verification (DV), CL-Proxy Signcryption Generation (CL-PSG), and CL-Proxy Un-Signcryption (CL-PU-S). The subsequent sub-phases further define the constructions of the seven algorithms listed above. The symbols used in the proposed scheme are listed in Table 1.

**Table 1.** Symbol table.

| S. No | Symbol | Descriptions |
|:---:|:---:|:---:|
| 1 | $NP_{KC}$ | The network provider that serves as KGC |
| 2 | $k_{KC}$ | The given security parameter to $NP_{KC}$ based on elliptic curve |
| 3 | $G_{KC}$ | A cyclic group of elliptic curve selected by the network provider |
| 4 | $\gamma_{KC}$ | It is the generator of a cyclic group $G_{KC}$ |
| 5 | $Y_m$ | Indicates the length of plaintext |
| 6 | $\left\| Z_{q_{KC}}{}^* \right\|$ | Indicates the length of selected parameter |
| 7 | $\Phi_{KC}$ | Indicates the master secret/master private key of $NP_{KC}$ |
| 8 | $\delta_{KC}$ | Indicates the master Public/Public key of $NP_{KC}$ |
| 9 | $PAR_{KC}$ | Represents the public parameter param that is distributed in a network |
| 10 | $PP_i$ | Represents the participated users, i.e., $(CC, GCS, drone)$ |
| 11 | $P_i$ | Represents the partial private key of participated users, i.e., $(CC, GCS, drone)$ |
| 12 | $V_{PP_i}$ | Represents the shared secret key between the participated users, i.e., $(CC, GCS, drone)$ and $NP_{KC}$ |
| 13 | $EID_{PP_i}$ | Represents the encrypted identity of participated users, i.e., $(CC, GCS, drone)$ |
| 14 | $ID_{PP_i}$ | Represents the identity of participated users, i.e., $(CC, GCS, drone)$ |
| 15 | $U_{PP_i}, X_{KC}$ | Represents the public key pair of participated users, i.e., $(CC, GCS, drone)$ |
| 16 | $P_i, \beta_{PP_i}$ | Represents the private key pair of participated users, i.e., $(CC, GCS, drone)$ |
| 17 | $U_{CC}, X_{CC}$ | Represents the public key pair of Control Centre $(CC)$ |
| 18 | $P_{CC}, \beta_{CC}$ | Represents the private key pair of Control Centre $(CC)$ |
| 19 | $U_{GCS}, X_{GCS}$ | Represents the public key pair of Ground Control Station $(GCS)$ |
| 20 | $U_{drone}, X_{drone}$ | Represents the public key pair of drone |
| 21 | $P_{drone}, \beta_{drone}$ | Represents the private key pair of drone |
| 22 | $P_{GCS}, \beta_{GCS}$ | Represenst the private key pair of Ground Control Station $(GCS)$ |
| 23 | $H_{KC1}, H_{KC2}, H_{KC3}, H_{KC4}$ | Represents secure cryptographic hash functions |
| 24 | $\oplus$ | It is used for encryption and decryptions |
| 25 | $S_{m_w}$ | It represents the signature generated on warrant message |
| 26 | $m_w$ | Represents the warrant message that contains the delegation durations |
| 27 | $C_{GCS}$ | Represents the ciphertext, which is generated by GCS |
| 28 | $K$ | Represents the shared secret key between GCS and drone |
| 29 | $m$ | Represents the plaintext, which is chosen by GCS |
| 30 | $S_{GCS}$ | Represents the signature generated on message |

**Setup:** Here, the network provider ($NP_{KC}$) assumes the role of KGC; when it receives the security parameter $k_{KC}$, $NP_{KC}$ executes the steps outlined below.

- Selects the group $G_{KC}$ of order $q_{KC}$ and $\gamma_{KC}$, which will be the generator of $G_{KC}$;
- Selects four hash functions $H_{KC1} : \{0,1\}^* \times G_{KC} \rightarrow Z_{q_{KC}}{}^*, H_{KC2} : \{0,1\}^* \times G_{KC} \rightarrow Z_{q_{KC}}{}^*, H_{KC3} : \{0,1\}^* \times G_{KC} \rightarrow Z_{q_{KC}}{}^*, H_{KC4} : \{0,1\}^* \times G_{KC} \rightarrow Z_{q_{KC}}{}^*$;
- Sets $Y_m$ is the plaintext length and $|Z_{q_{KC}}{}^*|$ will be the length of selected parameter;
- Selects the system private key as $\Phi_{KC} \in Z_{q_{KC}}{}^*$ and computes the public key $\delta_{KC} = \Phi_{KC}.\gamma_{KC}$;
- $NP_{KC}$ can made $PAR_{KC} = \{H_{KC1}, H_{KC2}, H_{KC3}, H_{KC4}, \delta_{KC}, \gamma_{KC}, G_{KC}, Z_{q_{KC}}{}^*\}$ as the public parameter and distributes it throughout a network.

**Partial Key Generation (PCGU):** If a participant $(PP_i)$, where $i = (CC, GCS, drone)$, desires a partial private key $(P_i)$ from $NP_{KC}$, it first selects $\beta_{PP_i} \in Z_{q_{KC}}{}^*$, computes $V_{PP_i} = \beta_{PP_i}.\delta_{KC}$, computes $U_{PP_i} = \beta_{PP_i}.\gamma_{KC}$, calculates $EID_{PP_i} = ID_{PP_i} \oplus V_{PP_i}$, and then, sends $(EID_{PP_i}, U_{PP_i})$ through insecure network to $NP_{KC}$. Alternatively, when $NP_{KC}$ receives $(EID_{PP_i}, U_{PP_i})$, it executes the following calculations: Computes $V_{PP_i} = \Phi_{KC}.U_{PP_i}$, recovers $PP_i$ identity as $ID_{PP_i} = EID_{PP_i} \oplus V_{PP_i}$, and then, $NP_{KC}$ selects $\alpha_{KC} \in Z_{q_{KC}}{}^*$ and computes $X_{KC} = \alpha_{KC}.\gamma_{KC}$. In addition, $NP_{KC}$ computes $P_i = \alpha_{KC} + \Phi_{KC}H_{KC1}(X_{KC}, ID_{PP_i})$, calculates $EP_i = (P_i, X_{KC}) \oplus V_{PP_i}$, and sends $EP_i$ as an encrypted partial private and public key to $PP_i$ over an insecure network.

**Public and Private Key Generation (PBCGU):** When $PP_i$ receives $EP_i$, it computes $(P_i, X_{KC}) = EP_i \oplus V_{PP_i}$, sets $(U_{PP_i}, X_{KC})$ to his public key and sets $(P_i, \beta_{PP_i})$ to his private key.

**Delegation Generation (DG):** This phase is run by the CC and when it receives $PAR_{KC} = \{H_{KC1}, H_{KC2}, H_{KC3}, H_{KC4}, \delta_{KC}, \gamma_{KC}, G_{KC}, Z_{q_{KC}}{}^*\}$, $(ID_{CC}, ID_{CG}, P_{CC}, \beta_{CC}, U_{CG}, X_{CG})$, where $ID_{CC}$ and $ID_{GCC}$ are the identities of CC and GCS, respectively, $(P_{CC}, \beta_{CC})$ is CC's private key pair, and $(U_{CG}, X_{CG})$ is GCS's public key pair.

Consequently, CC will execute the subsequent steps to generate a delegation signature for warrant $m_w$.

- It selects $A_{CC} \in Z_{q_{KC}}{}^*$, computes $O_{CC} = A_{CC}.\gamma_{KC}$, and $H_2 = H_{KC2}(U_{GCS}, U_{CC}, ID_{GCS}, ID_{CC}, O_{CC}, m_w)$;
- Computes $S_{m_w} = \frac{\beta_{CC} + A_{CC}}{H_2 + \beta_{CC} + P_{CC}} \, mod \, q$ and sends the triple $(m_w, S_{m_w}, O_{CC})$ as a delegation to the GCS through an open network.

**Delegation Verification (DV):** When $(m_w, S_{m_w}, O_{CC})$ is received by the GCS, the following verification procedures are carried out.

- Computes $H_2 = H_{KC2}(U_{GCS}, U_{CC}, ID_{GCS}, ID_{CC}, O_{CC}, m_w)$ and $H_1 = H_{KC1}(ID_{CC}, X_{CC})$;
- If $S_{m_w}(X_{CC} + U_{CC} + H_1.\delta_{KC} + H_2.\gamma_{KC}) = U_{CC} + O_{CC}$, then accept $(m_w, S_{m_w}, O_{CC})$; otherwise, an error message is returned.

**CL-Proxy Signcryption Generation (CL-PSG):** This phase is executed by the GCS, which generates a certificateless proxy signcryption using the procedures below.

- It selects $F_{GCS} \in Z_{q_{KC}}{}^*$, computes $Q_{GCS} = F_{GCS}.\gamma_{KC}$;
- Computes $H_3 = H_{KC3}(U_{GCS}, U_{CC}, ID_{GCS}, ID_{CC}, Q_{GCS}, m)$;
- Computes $H_1 = H_{KC1}(ID_{drone}, X_{drone})$ and $K = F_{GCS}(U_{drone} + X_{drone} + \delta_{KC}.H_1)$;
- Computes $C_{GCS} = H_{KC4}(K) \oplus m$ and $S_{GCS} = \frac{\beta_{GCS} + F_{GCS}}{H_3 + \beta_{GCS} + P_{GCS}} \, mod \, q$;
- Finally, it sends the triple $(C_{GCS}, S_{GCS}, Q_{GCS})$ as a proxy signcryption to *drone* via an open network.

**CL-Proxy Un-Signcryption (CL-PU-S):** When $(C_{GCS}, S_{GCS}, Q_{GCS})$ is received by the drone, it performs the following verifications steps.

- Computes $K = Q_{GCS}(\beta_{drone} + P_{drone})$ and $m = H_{KC4}(K) \oplus C_{GCS}$;
- Computes $H_3 = H_{KC3}(U_{GCS}, U_{CC}, ID_{GCS}, ID_{CC}, Q_{GCS}, m)$ and $H_1{}^\$ = H_{KC1}(ID_{GCS}, X_{GCS})$;
- If $S_{GCS}\left(X_{GCS} + U_{GCS} + H_1{}^\$.\delta_{KC} + H_3.\gamma_{KC}\right) = U_{GCS} + Q_{GCS}$, then accept $(C_{GCS}, S_{GCS}, Q_{GCS})$, otherwise, an error message is returned.

**Correctness**

The $NP_{KC}$ can compute the secret key $V_{PP_i}$ by using the following computations.

$$V_{PP_i} = \Phi_{KC}.U_{PP_i} = \Phi_{KC}.\beta_{PP_i}.\gamma_{KC} = \Phi_{KC}.\gamma_{KC}.\beta_{PP_i} = \delta_{KC}.\beta_{PP_i} = V_{PP_i}$$

The $NP_{KC}$ recovers/ decrypts the user identity by using the following computations.

$$ID_{PP_i} = EID_{PP_i} \oplus V_{PP_i} = EID_{PP_i} \oplus V_{PP_i} = ID_{PP_i} \oplus V_{PP_i} \oplus V_{PP_i} = ID_{PP_i}$$

The $PP_i$ recovers/decrypts the partial public and partial private key by using the following computations.

$$(P_i, X_{KC}) = EP_i \oplus V_{PP_i} = EP_i \oplus V_{PP_i} = (P_i, X_{KC}) \oplus V_{PP_i} \oplus V_{PP_i} = (P_i, X_{KC})$$

The *GCS* can verify the delegated text $(m_w, S_{m_w}, O_{CC})$ by using the following computations.

$$S_{m_w}(X_{CC} + U_{CC} + H_1.\delta_{KC} + H_2.\gamma_{KC}) = U_{CC} + O_{CC} = S_{m_w}(X_{CC} + U_{CC} + H_1.\delta_{KC} + H_2.\gamma_{KC}) =$$
$$\frac{\beta_{CC} + A_{CC}}{H_2 + \beta_{CC} + P_{CC}}(X_{CC} + U_{CC} + H_1.\delta_{KC} + H_2.\gamma_{KC}) = \frac{\beta_{CC} + A_{CC}}{H_2 + \beta_{CC} + P_{CC}}(\alpha_{CC}.\gamma_{KC} + U_{CC} + H_1.\delta_{KC} + H_2.\gamma_{KC})$$
$$= \frac{\beta_{CC} + A_{CC}}{H_2 + \beta_{CC} + P_{CC}}(\alpha_{CC}.\gamma_{KC} + \beta_{CC}.\gamma_{KC} + H_1.\delta_{KC} + H_2.\gamma_{KC}) =$$
$$\frac{\beta_{CC} + A_{CC}}{H_2 + \beta_{CC} + P_{CC}}(\alpha_{CC}.\gamma_{KC} + \beta_{CC}.\gamma_{KC} + H_1.\Phi_{KC}.\gamma_{KC} + H_2.\gamma_{KC}) =$$
$$\frac{\beta_{CC} + A_{CC}}{H_2 + \beta_{CC} + P_{CC}}(\alpha_{CC}.\gamma_{KC} + \beta_{CC}.\gamma_{KC} + H_{KC1}(ID_{CC}, X_{CC}).\Phi_{KC}.\gamma_{KC} + H_2.\gamma_{KC}) =$$
$$\frac{\beta_{CC} + A_{CC}}{H_2 + \beta_{CC} + P_{CC}}(\alpha_{CC} + \beta_{CC} + H_{KC1}(ID_{CC}, X_{CC}).\Phi_{KC} + H_2).\gamma_{KC} =$$
$$\frac{\beta_{CC} + A_{CC}}{H_2 + \beta_{CC} + P_{CC}}(\beta_{CC} + \alpha_{CC} + H_{KC1}.\Phi_{KC}.(ID_{CC}, X_{CC}) + H_2).\gamma_{KC} =$$
$$\frac{\beta_{CC} + A_{CC}}{H_2 + \beta_{CC} + P_{CC}}(\beta_{CC} + P_{CC} + H_2).\gamma_{KC} = (\beta_{CC} + A_{CC}).\gamma_{KC} = (\beta_{CC}.\gamma_{KC} + A_{CC}.\gamma_{KC}) = U_{CC} + O_{CC}$$

The drone can compute the secret key *K* by using the following computations.

$$K = Q_{GCS}(\beta_{drone} + P_{drone}) = Q_{GCS}(\beta_{drone} + P_{drone}) = F_{GCS}.\gamma_{KC}(\beta_{drone} + P_{drone}) =$$
$$F_{GCS}.\gamma_{KC}(\beta_{drone} + \alpha_{drone} + \Phi_{KC}H_{KC1}(X_{drone}, ID_{drone})) =$$
$$F_{GCS}(\beta_{drone}.\gamma_{KC} + \alpha_{drone}.\gamma_{KC} + \Phi_{KC}.\gamma_{KC}.H_{KC1}(X_{drone}, ID_{drone})) =$$
$$F_{GCS}(U_{drone} + \alpha_{drone}.\gamma_{KC} + \Phi_{KC}.\gamma_{KC}.H_{KC1}(X_{drone}, ID_{drone})) =$$
$$F_{GCS}(U_{drone} + X_{drone} + \Phi_{KC}.\gamma_{KC}.H_{KC1}(X_{drone}, ID_{drone})) =$$
$$F_{GCS}(U_{drone} + X_{drone} + \delta_{KC}.H_{KC1}(X_{drone}, ID_{drone})) = F_{GCS}(U_{drone} + X_{drone} + \delta_{KC}.H_1) = K$$

The drone can recover/decrypts the message *m* by using the following computations.

$$m = H_{KC4}(K) \oplus C_{GCS} = H_{KC4}(K) \oplus H_{KC4}(K) \oplus m = m$$

The drone can verify the proxy signcrypted text $(C_{GCS}, S_{GCS}, Q_{GCS})$ by using the following computations.

$$S_{GCS}\left(X_{GCS} + U_{GCS} + H_1{}^{\$}.\delta_{KC} + H_3.\gamma_{KC}\right) = U_{GCS} + Q_{GCS} =$$
$$S_{GCS}\left(X_{GCS} + U_{GCS} + H_1{}^{\$}.\delta_{KC} + H_3.\gamma_{KC}\right) = \frac{\beta_{GCS} + F_{GCS}}{H_3 + \beta_{GCS} + P_{GCS}}\left(X_{GCS} + U_{GCS} + H_1{}^{\$}.\delta_{KC} + H_3.\gamma_{KC}\right)$$
$$= \frac{\beta_{GCS} + F_{GCS}}{H_3 + \beta_{GCS} + P_{GCS}}\left(\alpha_{GCS}.\gamma_{KC} + U_{GCS} + H_1{}^{\$}.\delta_{KC} + H_3.\gamma_{KC}\right) =$$
$$\frac{\beta_{GCS} + F_{GCS}}{H_3 + \beta_{GCS} + P_{GCS}}\left(\alpha_{GCS}.\gamma_{KC} + \beta_{GCS}.\gamma_{KC} + H_1{}^{\$}.\delta_{KC} + H_3.\gamma_{KC}\right) =$$
$$\frac{\beta_{GCS} + F_{GCS}}{H_3 + \beta_{GCS} + P_{GCS}}\left(\alpha_{GCS}.\gamma_{KC} + \beta_{GCS}.\gamma_{KC} + H_1{}^{\$}.\Phi_{KC}.\gamma_{KC} + H_3.\gamma_{KC}\right) =$$
$$\frac{\beta_{GCS} + F_{GCS}}{H_3 + \beta_{GCS} + P_{GCS}}\left(\alpha_{GCS} + \beta_{GCS} + H_1{}^{\$}.\Phi_{KC} + H_3\right).\gamma_{KC} =$$
$$\frac{\beta_{GCS} + F_{GCS}}{H_3 + \beta_{GCS} + P_{GCS}}\left(\beta_{GCS} + \alpha_{GCS} + H_1{}^{\$}.\Phi_{KC} + H_3\right).\gamma_{KC} =$$
$$\frac{\beta_{GCS} + F_{GCS}}{H_3 + \beta_{GCS} + P_{GCS}}(\beta_{GCS} + \alpha_{GCS} + H_{KC1}(ID_{GCS}, X_{GCS}).\Phi_{KC} + H_3).\gamma_{KC} =$$
$$\frac{\beta_{GCS} + F_{GCS}}{H_3 + \beta_{GCS} + P_{GCS}}(\beta_{GCS} + \alpha_{GCS} + \Phi_{KC}.H_{KC1}(ID_{GCS}, X_{GCS}) + H_3).\gamma_{KC} =$$
$$\frac{\beta_{GCS} + F_{GCS}}{H_3 + \beta_{GCS} + P_{GCS}}(\beta_{GCS} + P_{GCS} + H_3).\gamma_{KC} = (\beta_{GCS} + F_{GCS})\gamma_{KC} = (U_{GCS} + Q_{GCS})$$

## 5. Security Analysis

In order to carry out the provable security analysis of the proposed scheme, which makes use of a well-known method of formal security analysis called as the random or-

acle model, the proposed scheme is secured against Type 1 ($NP_{A_1}$) and Type 2 ($NP_{A_2}$) adversaries. When these adversaries attempted to violate the confidentiality and forge the original signature, the subsequent sub-steps elucidated the role of (Type 1 ($NP_{A_1}$) and Type 2 ($NP_{A_2}$) and the security hard problems upon which our scheme's security is based.

**Elliptic Curve Diffie–Hellman Problem (ECDHP):** Given ($\gamma_{KC}, a.\gamma_{KC}, b.\gamma_{KC}$), finding the values of $a, b$ from $a.\gamma_{KC}, b.\gamma_{KC}$ is hard and is reported to ECDHP.

**Elliptic Curve Discrete Logarithm Problem (ECDLP):** Given ($\gamma_{KC}, a.\gamma_{KC}$), finding the value of $a$ from $a.\gamma_{KC}$ is hard and is reported to ECDLP.

By utilizing the following theorems, we will elucidate how the proposed scheme withstands against Type 1 ($NP_{A_1}$) and Type 2 ($NP_{A_2}$).

**Theorem 1.** *In this theorem, we are going to perform the IND-SFCPS-CCA2 game between $NP_{A_1}$ and $NP_{CR}$ to break the confidentiality of the proposed scheme, in which $NP_{CR}$ performs the role helper for $NP_{A_1}$ to obtain the solution of ECDHP. Suppose $NP_{A_1}$ wins with the non-ignorable advantage ($AD_{A_1}$) in the game IND-SFCPS-CCA2 and $NP_{CR}$ get the solution for ECDHP with the advantage of $AD_{A_1}{}^{IND-SFCPS-CCA2} \geq \frac{AD_{A_1}}{q_1{}^2 q_4}\left(1 - \frac{1}{q_{PS}+1}\right)q_{PS}\frac{1}{q_{PS}+1}$. Where, $q_1$ and $q_4$ represent a query for $H_{KC1}$ and $H_{KC4}$, and $q_{PS}$ represents a proxy signcryption query.*

**Proof.** Given ($\gamma_{KC}, a.\gamma_{KC}, b.\gamma_{KC}$)**,** the task of $NP_{A_1}$ is to extract the value $a, b$ from $a.\gamma_{KC}$, $b.\gamma_{KC}$ with the help of $NP_{CR}$. The following is the process in which $NP_{A_1}$ with the help of $NP_{CR}$ could solve the above problem.

**Setup:** Here, $NP_{CR}$ selects $\Phi_{KC}{}^* \in Z_{q_{KC}}{}^*$, computes $\delta_{KC}$, makes a param $PAR_{KC}$, and sends $PAR_{KC}$ to $NP_{A_1}$. Then, $NP_{A_1}$ can ask for the following queries.

*Find Stage:* Here, in this section, $NP_{A_1}$ can ask for the following polynomial bounded queries.

$H_{KC1}$ *Query:* If $NP_{CR}$ receives ($X_j, ID_j$) as a query from $NP_{A_1}$, $NP_{CR}$ checks for ($X_j, ID_j$, $H_1, l$) in the list $L_{H_{KC1}}$, if it is available, it sends $H_1$ to $NP_{A_1}$; otherwise, $NP_{CR}$ choose $l \in \{0,1\}$, here, its probability as $\Pr(l = 1) = \frac{1}{q_{PS}+1}$. Then, it checks, if ($l = 0$), and then chooses $H_1 \in Z_{q_{KC}}{}^*$, sends $H_1$ to $NP_{A_1}$ and adds ($X_j, ID_j, H_1, l$) into $L_{H_{KC1}}$. If ($l = 1$), $NP_{CR}$ sets $k_{KC} = H_1$, and returns $k_{KC}$ to $NP_{A_1}$.

$H_{KC2}$ *Query:* If $NP_{CR}$ receives ($U_j, ID_j, O_j, m_w$) as a query from $NP_{A_1}$, checks for ($U_j, ID_j$, $O_j, m_w, H_2$) in the list $L_{H_{KC2}}$, if it is available, it sends $H_2$ to $NP_{A_1}$; otherwise, $NP_{CR}$ chooses $H_2 \in Z_{q_{KC}}{}^*$, sends $H_2$ to $NP_{A_1}$ and adds ($U_j, ID_j, O_j, m_w, H_2$) into $L_{H_{KC1}}$.

$H_{KC3}$ *Query:* If $NP_{CR}$ receives ($U_j, ID_j, O_j, m$) as a query from $NP_{A_1}$, $NP_{CR}$ checks for ($U_j, ID_j, O_j, m, H_3$) in the list $L_{H_{KC3}}$, if it is available, it sends $H_3$ to $NP_{A_1}$; otherwise, $NP_{CR}$ chooses $H_3 \in Z_{q_{KC}}{}^*$, sends $H_3$ to $NP_{A_1}$ and adds ($U_j, ID_j, O_j, m, H_3$) into $L_{H_{KC3}}$.

$H_{KC4}$ *Query:* If $NP_{CR}$ receives ($ID_j, K$) as a query from $NP_{A_1}$, $NP_{CR}$ checks for ($ID_j, K, H_4$) in the list $L_{H_{KC4}}$, if it is available, it sends $H_4$ to $NP_{A_1}$; otherwise, $NP_{CR}$ chooses $H_4 \in Z_{q_{KC}}{}^*$, sends $H_4$ to $NP_{A_1}$ and adds ($ID_j, K, H_4$) into $L_{H_{KC4}}$.

*PCGU Query:* If $NP_{CR}$ receives ($X_j, ID_j, P_j$) as a query from $NP_{A_1}$, $NP_{CR}$ checks for ($X_j$, $ID_j, P_j$) in the list $L_{PCGU}$, if it is available, it sends ($X_j, P_j$) to $NP_{A_1}$; otherwise, $NP_{CR}$ chooses $\alpha_j, \Phi_j \in Z_{q_{KC}}{}^*$, computes $P_j = \alpha_j + \Phi_j H_{KC1}(X_j, ID_j)$, sends ($X_j, P_j$) to $NP_{A_1}$, and adds ($X_j, ID_j, P_j$) into $L_{PCGU}$.

*Private Key Query:* If $NP_{CR}$ receives ($\beta_j, ID_j, P_j$) as a query from $NP_{A_1}$, $NP_{CR}$ checks for ($\beta_j, ID_j, P_j$) in the list $L_{PKQ}$, if it is available, it sends ($\beta_j, P_j$) to $NP_{A_1}$. Otherwise, $NP_{CR}$ chooses $\beta_j \in Z_{q_{KC}}{}^*$, obtained $P_j$ from *PCGU Query*, sends ($\beta_j, P_j$) to $NP_{A_1}$, and add ($\beta_j, ID_j, P_j$) into $L_{PKQ}$.

*Public Key Query:* If $NP_{CR}$ receives ($X_j, ID_j, U_j$) as a query from $NP_{A_1}$, $NP_{CR}$ checks for ($X_j, ID_j, U_j$) in the list $L_{PBKQ}$, if it is available, it sends ($X_j, U_j$)) to $NP_{A_1}$. Otherwise, $NP_{CR}$ searches and finds ($\beta_j, X_j$) from $L_{PKQ}$ and $L_{PCGU}$, and then computes $U_j = \beta_j.\gamma_{KC}$, sends ($X_j, U_j$)) to $NP_{A_1}$, and adds ($X_j, ID_j, U_j$) into $L_{PBKQ}$.

*Replace Public Key Query:* $NP_{A_1}$ sends ($X_j{}', U_j{}'$) to $NP_{CR}$ and can replace ($X_j, U_j$) on ($X_j{}', U_j{}'$) for the identity $ID_j$.

*Delegation Generation Query:* $NP_{A_1}$ sends two identity ($ID_{CC}, ID_{GCS}$) and a warrant $m_w$ to $NP_{CR}$, it then checks the tuple ($X_{CC}, ID_{CC}$) in $L_{H_{KC1}}$. If ($l = 1$), it can abort further processing. Otherwise, it extracts ($\beta_{CC}, P_{CC}$) from $L_{PKQ}$, $H_2$ from $L_{H_{KC2}}$, chooses $A_{CC} \in Z_{q_{KC}}^*$, generates ($m_w, S_{CC}, O_{CC}$), and sends it to $NP_{A_1}$.

*CL-Proxy Signcryption Query:* $NP_{A_1}$ sends two identities ($ID_{drone}, ID_{GCS}$) and a message ($m$) to $NP_{CR}$, it then checks the tuple ($X_{GCS}, ID_{GCS}$) in $L_{H_{KC1}}$; if ($l = 1$), it can abort further processing. Otherwise, it extracts ($\beta_{GCS}, P_{GCS}$) from $L_{PKQ}$, $H_2$ from $L_{H_{KC2}}$, chooses $F_{GCS} \in Z_{q_{KC}}^*$, generates ($C_{GCS}, S_{GCS}, Q_{GCS}$), and sends it to $NP_{A_1}$.

*CL-Proxy Un-Signcryption Query:* $NP_{A_1}$ sends two identities ($ID_{drone}, ID_{GCS}$) and ($C_{GCS}, S_{GCS}, Q_{GCS}$) to $NP_{CR}$, it then checks the tuple ($X_{drone}, ID_{drone}$) in $L_{H_{KC1}}$, the response is then provided in the subsequent methods.

1.  If ($l = 0$), $NP_{CR}$ can obtain ($X_{GCS}, ID_{GCS}, U_{GCS}$) from $L_{PBKQ}$ according to identity $ID_{GCS}$, ($\beta_{drone}, ID_{drone}, drone$) from $L_{PKQ}$, performs the *Proxy Un − Signcryption* algorithm and sends ($m$) to $NP_{A_1}$.

2.  If ($l = 1$), $NP_{CR}$ can get ($H_4$) from $L_{H_{KC4}}$ and computes $m = H_{KC4}(K) \oplus C_{GCS}$ perform the *Proxy Un − Signcryption* algorithm. $NP_{CR}$ can further obtains ($X_{GCS}, ID_{GCS}, H_1^{\$}, l$) from $L_{H_{KC1}}$, ($X_{GCS}, U_{GCS}$) from $L_{PBKQ}$, ($H_3$) from the list $L_{H_{KC3}}$, and $NP_{CR}$ can verify the equation $S_{GCS}\left(X_{GCS} + U_{GCS} + H_1^{\$}.\delta_{KC} + H_3.\gamma_{KC}\right) = U_{GCS} + Q_{GCS}$. If the condition is met, the output is (m); otherwise, the procedure is repeated with new parameters.

*Challenge Stage:* Suppose $m_{KC1}$ and $m_{KC2}$ is adaptively generated two distinct messages by $NP_{A_1}$ and sends ($m_{KC1}, m_{KC2}$) and two challenged identities ($ID_{GCS}, ID_{UAV}$) to $NP_{CR}$. Then, $NP_{CR}$ checks for the tuple ($X_{drone}, ID_{drone}$) in $L_{H_{KC1}}$, if ($l = 0$), $NP_{CR}$ stop; otherwise, it chooses $\left(C_{GCS}', S_{GCS}', Q_{GCS}'\right) \in Z_{q_{KC}}^*$ randomly and sends it to $NP_{A_1}$ as a challenge ciphertext.

*Guess Stage:* $NP_{A_1}$ can make sure $H_{KC1}$ *Query*, $H_{KC2}$ *Query*, $H_{KC3}$ *Query*, $H_{KC4}$ *Query*, *PCGU Query*, *Private Key Query*, *Public Key Query*, *Replace Public Key Query*, *Delegation Generation Query*, *CL-Proxy Signcryption Query*, *CL-Proxy Un-Signcryption Query* is performed as same as above in Find Stage. So, $NP_{CR}$ returns $l'$, $NP_{A_1}$ can made $H_{KC4}$ *Query* with $K' = F_{GCS}(U_{drone} + X_{drone} + \delta_{KC}.H_1)$. In this situation, the valid answer for ECDHP is included to $L_{H_{KC4}}$. The second situation is that $NP_{CR}$ can ignore the randomly selected/guessed value of $NP_{A_1}$, then $NP_{CR}$ randomly selects $K'$ from $L_{H_{KC4}}$ and computes $\left(\frac{K' - (\beta_{drone} + \alpha_{drone})Q_{GCS}'}{k_{KC}}\right) = F_{GCS}.\Phi_{KC}.\gamma_{KC}$, where $NP_{CR}$ already knows the value $\beta_{drone}, \alpha_{drone}, Q_{GCS}'$, and $K'$. Otherwise, $NP_{CR}$ failed to solve ECDHP.

So, we are going to evaluate the above process with success probability. The success probability will be $\frac{1}{q_1^2}$ when $NP_{A_1}$ made *PCGU Query* and *Private Key Query* for $ID_{drone}$. The success probability will be $\frac{1}{q_4}$ when $NP_{CR}$ successfully selects $K'$ from $L_{H_{KC4}}$. The success probability will $\frac{AD_{A_1}}{q_1^2 q_4}\left(1 - \frac{1}{q_{PS}+1}\right)q_{PS}\frac{1}{q_{PS}+1}$ when $NP_{CR}$ is not halting this game's simulation. We can say that $NP_{CR}$ can obtain the solution for ECDHP with the advantage as follows: $AD_{A_1}^{IND-SFCPS-CCA2} \geq \frac{AD_{A_1}}{q_1^2 q_4}\left(1 - \frac{1}{q_{PS}+1}\right)q_{PS}\frac{1}{q_{PS}+1}.$ $\square$

**Theorem 2.** *In this theorem, we are going to perform the IND-SFCPS-CCA2 game between $NP_{A_2}$ and $NP_{CR}$ to breaks the confidentiality of the proposed Scheme, in which $NP_{CR}$ performs the role of helper for $NP_{A_2}$ to obtain the solution of ECDHP. Suppose $NP_{A_2}$ wins with the non-ignorable advantage ($AD_{A_2}$) in the game IND-SFCPS-CCA2 and $NP_{CR}$ gets the solution for ECDHP with the advantage of $AD_{A_2}^{IND-SFCPS-CCA2} \geq \frac{AD_{A_2}}{q_1^2 q_4}\left(1 - \frac{1}{q_{PS}+1}\right)q_{PS}\frac{1}{q_{PS}+1}$. Where $q_1$ and $q_4$ represents a query for $H_{KC1}$ and $H_{KC4}$, and $q_{PS}$ represents a proxy signcryption query.*

**Proof.** Given ($\gamma_{KC}, a.\gamma_{KC}, b.\gamma_{KC}$), the task of $NP_{A_2}$ is to extract the value $a, b$ from $a.\gamma_{KC}$, $b.\gamma_{KC}$ with the help of $NP_{CR}$. The following is the process in which $NP_{A_2}$ with the help of

$NP_{CR}$ could solve the above problem.

***Setup:*** Here, $NP_{CR}$ selects $\Phi_{KC} \in Z_{q_{KC}}{}^*$, computes $\delta_{KC}$, make a param $PAR_{KC}$, and sends $PAR_{KC}$ and $\Phi_{KC}$ to $NP_{A_2}$. Then, $NP_{A_2}$ can ask for the following queries.

***Find Stage:*** Here, in this section, $NP_{A_2}$ can ask for the following polynomial bounded queries.

The queries such as $H_{KC1}$ *Query, $H_{KC2}$ Query, $H_{KC3}$ Query, $H_{KC4}$ Query* are identical to those performed in Theorem 1.

***PCGU Query***: If $NP_{CR}$ receives $(X_j, ID_j, P_j)$ as a query from $NP_{A_1}$, $NP_{CR}$ checks for $(X_j, ID_j, P_j)$ in the list $L_{PCGU}$. If it is available, it sends $(X_j, P_j)$ to $NP_{A_2}$. Otherwise, $NP_{CR}$ chooses $\alpha_j, \Phi_j \in Z_{q_{KC}}{}^*$, computes $P_j = \alpha_j + \Phi_j H_{KC1}(X_j, ID_j)$, sends $(X_j, P_j)$ to $NP_{A_2}$, and adds $(X_j, ID_j, P_j)$ into $L_{PCGU}$.

***Private Key Query***: If $NP_{CR}$ receives $(\beta_j, ID_j, P_j)$ as a query from $NP_{A_2}$, $NP_{CR}$ checks for $(\beta_j, ID_j, P_j)$ in the list $L_{PKQ}$. If it is available, it sends $(\beta_j, P_j)$ to $NP_{A_2}$. Otherwise, $NP_{CR}$ chooses $\beta_j \in Z_{q_{KC}}{}^*$, obtained $P_j$ from *PCGU Query*, sends $(\beta_j, P_j)$ to $NP_{A_2}$, and adds $(\beta_j, ID_j, P_j)$ into $L_{PKQ}$.

***Public Key Query***: If $NP_{CR}$ receives $(X_j, ID_j, U_j)$ as a query from $NP_{A_2}$, $NP_{CR}$ checks for $(X_j, ID_j, U_j)$ in the list $L_{PBKQ}$, if it is available, it sends $(X_j, U_j))$ to $NP_{A_1}$. Otherwise, $NP_{CR}$ searches and finds $(\beta_j, X_j)$ from $L_{PKQ}$ and $L_{PCGU}$, and then computes $X_j = \Phi_{KC}.\gamma_{KC}$, sends $(X_j, U_j)$ to $NP_{A_2}$ and adds $(X_j, ID_j, U_j)$ into $L_{PBKQ}$.

***Delegation Generation Query:*** $NP_{A_2}$ sends two identity $(ID_{CC}, ID_{GCS})$ and a warrant $m_w$ to $NP_{CR}$, it then checks the tuple $(X_{CC}, ID_{CC})$ in $L_{H_{KC1}}$; if $(l = 1)$, it can abort further processing. Otherwise, it extracts $(\beta_{CC}, P_{CC})$ from $L_{PKQ}$, $H_2$ from $L_{H_{KC2}}$, chooses $A_{CC} \in Z_{q_{KC}}{}^*$, generates $(m_w, S_{CC}, O_{CC})$, and sends it to $NP_{A_2}$.

***CL-Proxy Signcryption Query:*** $NP_{A_2}$ sends two identities $(ID_{drone}, ID_{GCS})$ and a message $(m)$ to $NP_{CR}$, it then checks the tuple $(X_{GCS}, ID_{GCS})$ in $L_{H_{KC1}}$; if $(l = 1)$, it can abort further processing. Otherwise, it extracts $(\beta_{GCS}, P_{GCS})$ from $L_{PKQ}$, $H_2$ from $L_{H_{KC2}}$, chooses $F_{GCS} \in Z_{q_{KC}}{}^*$, generates $(C_{GCS}, S_{GCS}, Q_{GCS})$, and sends it to $NP_{A_2}$.

***CL-Proxy Un-Signcryption Query:*** $NP_{A_2}$ sends two identities $(ID_{drone}, ID_{GCS})$ and $(C_{GCS}, S_{GCS}, Q_{GCS})$ to $NP_{CR}$, it then checks the tuple $(X_{drone}, ID_{drone})$ in $L_{H_{KC1}}$, and it gives the response in the following ways.

1.　If $(l = 0)$, $NP_{CR}$ can obtain $(X_{GCS}, ID_{GCS}, U_{GCS})$ from $L_{PBKQ}$ according to identity $ID_{GCS}$, $(\beta_{drone}, ID_{drone}, P_{drone})$ from $L_{PKQ}$, perform the *Proxy Un − Signcryption* algorithm, and sends $(m)$ to $NP_{A_2}$.

2.　If $(l = 1)$, $NP_{CR}$ can obtain $(H_4)$ from $L_{H_{KC4}}$, compute $m = H_{KC4}(K) \oplus C_{GCS}$, and perform the *Proxy Un − Signcryption* algorithm. $NP_{CR}$ further can get $(X_{GCS}, ID_{GCS}, H_1{}^\$, l)$ from $L_{H_{KC1}}$, $(X_{GCS}, U_{GCS})$ from $L_{PBKQ}$, $(H_3)$ from the list $L_{H_{KC3}}$, and $NP_{CR}$ can verify the equation $S_{GCS}\left(X_{GCS} + U_{GCS} + H_1{}^\$.\delta_{KC} + H_3.\gamma_{KC}\right) = U_{GCS} + Q_{GCS}$; if it is satisfied, its output will be $(m)$, otherwise, it repeats this process again with new parameters.

***Challenge Stage:*** Suppose $m_{KC1}$ and $m_{KC2}$ adaptively generated two distinct messages by $NP_{A_2}$ and send $(m_{KC1}, m_{KC2})$ and two challenged identities $(ID_{GCS}, ID_{drone})$ to $NP_{CR}$. Then, $NP_{CR}$ checks for the tuple $(X_{drone}, ID_{drone})$ in $L_{H_{KC1}}$, if $(l = 0)$, $NP_{CR}$ stop; otherwise, it chooses $\left(C_{GCS}{}', S_{GCS}{}', Q_{GCS}{}'\right) \in Z_{q_{KC}}{}^*$ randomly and sends it to $NP_{A_2}$ as a challenge ciphertext.

***Guess Stage***: $NP_{A_2}$ can ensure $H_{KC1}$ *Query, $H_{KC2}$ Query, $H_{KC3}$ Query, $H_{KC4}$ Query, PCGU Query, Private Key Query, Delegation Generation Query, CL-Proxy Signcryption Query, CL-Proxy Un-Signcryption Query* is performed as same as above in *Find Stage* of *Theorem 1 and Public Key Query of Theorem 2*. So, $NP_{CR}$ returns $l'$, $NP_{A_2}$ can make $H_{KC4}$ *Query* with $K' = F_{GCS}(U_{drone} + X_{drone} + \delta_{KC}.H_1)$; in this situation, the valid answer for ECDHP includes $L_{H_{KC4}}$. The second situation is that $NP_{CR}$ can ignore the randomly selected/guessed value of $NP_{A_1}$, $NP_{CR}$ then randomly selects $K'$ from $L_{H_{KC4}}$ and computes $(K' - (\beta_{drone} + \Phi_{KC}k_{KC})Q_{GCS}{}') = F_{GCS}.\Phi_{KC}.\gamma_{KC}$, where $NP_{CR}$ already knows the value $\beta_{drone}, \alpha_{drone}, Q_{GCS}{}'$, and $K'$. Otherwise, $NP_{CR}$ failed to solve ECDHP.

Therefore, we will evaluate the preceding procedure based on its success probability. The success probability will be $\frac{1}{q_1^2}$ when $NP_{A_2}$ made *PCGU Query and Private Key Query* for $ID_{UAV}$. The success probability will be $\frac{1}{q_4}$ when $NP_{CR}$ successfully selects $K'$ from $L_{H_{KC4}}$. The success probability will $\frac{AD_{A_2}}{q_1^2 q_4}\left(1 - \frac{1}{q_{PS}+1}\right)q_{PS}\frac{1}{q_{PS}+1}$ when $NP_{CR}$ is not stopped in the simulation of this game. So, we can say that $NP_{CR}$ can obtain the solution for ECDHP with the following advantages: $AD_{A_2}{}^{IND-SFCPS-CCA2} \geq \frac{AD_{A_2}}{q_1^2 q_4}\left(1 - \frac{1}{q_{PS}+1}\right)q_{PS}\frac{1}{q_{PS}+1}$. $\square$

**Theorem 3.** *In this theorem, we are going to perform the EUF-SFCPS-CMA game between $NP_{A_1}$ and $NP_{CR}$ to forge the signature of the proposed scheme, in which $NP_{CR}$ perform the role helper for $NP_{A_1}$ to get the solution of ECDLP. Suppose $NP_{A_1}$ wins with the non-ignorable advantage ($AD_{A_1}$) in the game EUF-SFCPS-CMA and $NP_{CR}$ get the solution for ECDLP with the advantage of $AD_{A_1}{}^{IND-SFCPS-CCA2} \geq \frac{AD_{A_1}}{q_1^2}\left(1 - \frac{1}{q_{PS}+1}\right)q_{PS}$. Where $q_1$ and $q_4$ represents a query for $H_{KC1}$ and $H_{KC4}$, and $q_{PS}$ represents a proxy Signcryption query.*

**Proof.** Given $(\gamma_{KC}, a.\gamma_{KC})$, the task of $NP_{A_1}$ is to extract the value $a$ from $a.\gamma_{KC}$ with the help of $NP_{CR}$. The following are the processes in which $NP_{A_1}$ with the help of $NP_{CR}$, could solve the above problem.
*Setup:* Here, $NP_{CR}$ selects $\Phi_{KC}{}^* \in Z_{q_{KC}}{}^*$, computes $\delta_{KC}$, makes a param $PAR_{KC}$, and sends $PAR_{KC}$ to $NP_{A_1}$. Then, $NP_{A_1}$ can ask for the following queries.
*Find Stage:* Here, in this section, $NP_{A_1}$ can ask for the following polynomial bounded queries.
$H_{KC1}$ *Query*, $H_{KC2}$ *Query*, $H_{KC3}$ *Query*, $H_{KC4}$ *Query*, *PCGU Query*, *Private Key Query*, *Public Key Query*, *Replace Public Key Query*, *Delegation Generation Query*, *CL-Proxy Signcryption Query*, *CL-Proxy Un-Signcryption Query* is performed in the same way as above in *Find Stage of Theorem 1.*
*Forgery:* As $NP_{A_1}$ can ask for the following polynomial-bounded queries: $H_{KC1}$ *Query*, $H_{KC2}$ *Query*, $H_{KC3}$ *Query*, $H_{KC4}$ *Query*, *PCGU Query*, *Private Key Query*, *Public Key Query*, *Replace Public Key Query*, *Delegation Generation Query*, *CL-Proxy Signcryption Query*, *CL-Proxy Un-Signcryption Query* is performed as same as above in *Find Stage of Theorem 1* and generates a forged proxy signcryption triple $(C_{GCS}{}', S_{GCS}{}', Q_{GCS}{}')$ with the help of $NP_{CR}$. Note that $NP_{CR}$ can only solve the ECDLP if it accessed the actual value for $\beta_{CC}$ and $A_{CC}$ from $U_{CC} = \beta_{CC}.\gamma_{KC} = a.\gamma_{KC}$ and $O_{CC} = A_{CC}.\gamma_{KC} = a.\gamma_{KC}$.
So, we are going to evaluate the above process with success probability. The success probability will be $\frac{1}{q_1^2}$ when $NP_{A_1}$ made *PCGU Query and Private Key Query* for $ID_{UAV}$. The success probability will be $\frac{1}{q_4}$ when $NP_{CR}$ successfully selects $K'$ from $L_{H_{KC4}}$. The success probability will $\frac{AD_{A_1}}{q_1^2}\left(1 - \frac{1}{q_{PS}+1}\right)q_{PS}$ when $NP_{CR}$ does not stop the simulation of this game. So, we can say that $NP_{CR}$ can obtain solution for ECDHP with the followed advantages: $AD_{A_1}{}^{IND-SFCPS-CCA2} \geq \frac{AD_{A_1}}{q_1^2}\frac{AD_{A_1}}{q_1^2}\left(1 - \frac{1}{q_{PS}+1}\right)q_{PS}$. $\square$

**Theorem 4.** *In this theorem, we are going to perform the EUF-SFCPS-CMA game between $NP_{A_2}$ and $NP_{CR}$ to forge the signature of the proposed scheme, in which $NP_{CR}$ performs the role helper for $NP_{A_1}$ to get the solution of ECDHP. Suppose $NP_{A_2}$ wins with the non-ignorable advantage ($AD_{A_2}$) in the game EUF-SFCPS-CMA and $NP_{CR}$ get the solution for ECDLP with the advantage of $AD_{A_2}{}^{IND-SFCPS-CCA2} \geq \frac{AD_{A_2}}{q_1^2}\left(1 - \frac{1}{q_{PS}+1}\right)q_{PS}$, where $q_1$ and $q_4$ represents a query for $H_{KC1}$ and $H_{KC4}$, and $q_{PS}$ represents a proxy signcryption query.*

**Proof.** Given $(\gamma_{KC}, a.\gamma_{KC})$, the task of $NP_{A_2}$ is to extract the value $a$ from $a.\gamma_{KC}$ with the help of $NP_{CR}$. The following are the processes in which $NP_{A_2}$ with the help of $NP_{CR}$, could solve the above problem.
*Setup:* Here, $NP_{CR}$ selects $\Phi_{KC}{}^* \in Z_{q_{KC}}{}^*$, computes $\delta_{KC}$, makes a param $PAR_{KC}$, and sends

$PAR_{KC}$ and $\Phi_{KC}$ to $NP_{A_1}$. Then, $NP_{A_2}$ can ask for the following queries.

***Find Stage:*** Here, in this section, $NP_{A_2}$ can ask for the following polynomial-bounded queries: $H_{KC1}$ *Query*, $H_{KC2}$ *Query*, $H_{KC3}$ *Query*, $H_{KC4}$ *Query*, *PCGU Query*, *Private Key Query*, *Delegation Generation Query, CL-Proxy Signcryption Query, CL-Proxy Un-Signcryption Query* is performed in the same way as above in *Find Stage* of *Theorem 1 and Public Key Query of Theorem 2.*

***Forgery:*** As $NP_{A_2}$ can ask for the following polynomial-bounded queries: $H_{KC1}$ *Query*, $H_{KC2}$ *Query*, $H_{KC3}$ *Query*, $H_{KC4}$ *Query*, *PCGU Query*, *Private Key Query*, *Delegation Generation Query, CL-Proxy Signcryption Query, CL-Proxy Un-Signcryption Query* is performed as same as above in *Find Stage* of *Theorem 1 and Public Key Query of Theorem 2.* Furthermore, it generates a forged proxy signcryption triple $\left(C_{GCS}{}', S_{GCS}{}', Q_{GCS}{}'\right)$ with the help of $NP_{CR}$. Note that $NP_{CR}$ can only solve the ECDLP if it accessed the actual value for $\beta_{CC}$ and $A_{CC}$ from $U_{CC} = \beta_{CC}.\gamma_{KC} = a.\gamma_{KC}$ and $O_{CC} = A_{CC}.\gamma_{KC} = a.\gamma_{KC}$.

So, we are going to evaluate the above process with success probability. The success probability will be $\frac{1}{q_1{}^2}$ when $NP_{A_2}$ made *PCGU Query and Private Key Query* for $ID_{UAV}$. The success probability will be $\frac{1}{q_4}$ when $NP_{CR}$ successfully selects $K'$ from $L_{H_{KC4}}$. The success probability will $\frac{AD_{A_2}}{q_1{}^2}\left(1 - \frac{1}{q_{PS}+1}\right)q_{PS}$ when $NP_{CR}$ does not stop the simulation of this game. So, we can say that $NP_{CR}$ can obtain solution for ECDHP with the following advantages: $AD_{A_2}{}^{IND-SFCPS-CCA2} \geq \frac{AD_{A_2}}{q_1{}^2}\left(1 - \frac{1}{q_{PS}+1}\right)q_{PS}. \ \square$

## 6. Performance Comparison

This section is devoted to the performance comparison of the proposed scheme with existing equivalents schemes, such as those of which were proposed by Yanfeng et al. [25], Bhatia and Verma [26], Li et al. [27], and Qu and Zeng [28] in terms of computation and communication costs. The proposed scheme is presented in a clear and organized manner through figures and tables, which will help to better understand its viability.

### 6.1. Computational Cost

Tables 2 and 3 present a comparison of the computational cost. The tables present a performance comparison of the proposed scheme and the methods introduced by Yanfeng et al. [25], Bhatia and Verma [26], Li et al. [27], and Qu and Zeng [28], based on computation cost expressed in major operations and in milliseconds. The computation cost was evaluated utilizing the Raspberry board. Despite the availability of alternative replacements for RPI, which boast advanced hardware configurations such as LattePanda 4 G/64 GB, Qualcomm Dragon board, ODROID-XU4, and ASUS Tinker Board, among others, RPI remains widely regarded as the most economical and power-efficient choice. Additional compelling attributes of the RPI 4 that reinforce its choice include its integrated wireless network capabilities; specifically, dual-band 802.11 b/g/n/ac Wi-Fi and Bluetooth 5.0 BLE. In the present scheme implementation, the model and hardware specifications were delineated as follows: the Raspberry PI 4B (2019) is equipped with a 64-bit CPU architecture and a 1.5 GHz quad-core processor. It operates on the Ubuntu 20.04.2 LTS operating system and has a memory capacity of 8 GB, as reported in reference [29]. $NP_{ESM}$ represents elliptic curve scalar multiplications and $NP_{POP}$ represents pairing operation. It was observed, with respect to average time, that a single $NP_{ESM}$ takes 2.848 ms and $NP_{POP}$ takes 18.294 ms [29].

As shown in Figure 2, it was demonstrated that the proposed scheme had a lower computation cost in comparison to the extant schemes proposed by Yanfeng et al. [25], Li et al. [27], and Qu and Zeng [28]. The computation cost of the proposed scheme and that of Bhatia and Verma [18]'s scheme were identical; however, Bhatia and Verma [26]'s scheme had several serious flaws, such as requiring a secure channel for the distributions of partial private key, sending the identity in an open channel to $NP_{KC}$, which can compro-

mise the anonymity of the sender and receiver, and being susceptible if a Type 1 adversary replaced the user public key.

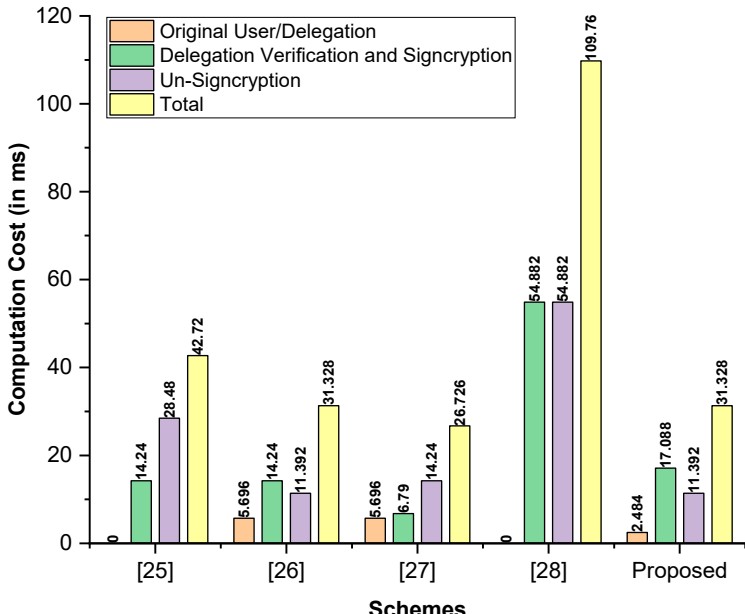

**Figure 2.** Comparison of the computational cost (in ms) of the proposed scheme and those proposed by Yanfeng et al. [25], Li et al. [27], and Qu and Zeng [28].

**Table 2.** Comparison of computation cost with major operations.

| Schemes | Original User/Delegation | Delegation Verification and Signcryption | Un-Signcryption | Total |
|---|---|---|---|---|
| Yanfeng et al. [25] | - | $5NP_{ESM}$ | $10NP_{ESM}$ | $15NP_{ESM}$ |
| Bhatia and Verma [26] | $2NP_{ESM}$ | $5NP_{ESM}$ | $4NP_{ESM}$ | $11NP_{ESM}$ |
| Li et al. [27] | $2NP_{ESM}$ | $7NP_{ESM}$ | $5NP_{ESM}$ | $14NP_{ESM}$ |
| Qu and Zeng [28] | — | $3NP_{POP}$ | $3NP_{POP}$ | $6NP_{POP}$ |
| Proposed Scheme | $1NP_{ESM}$ | $6NP_{ESM}$ | $4NP_{ESM}$ | $11NP_{ESM}$ |

**Table 3.** Comparison of computation cost (in ms).

| Schemes | Original User/Delegation | Delegation Verification and Signcryption | Un-Signcryption | Total |
|---|---|---|---|---|
| Yanfeng et al. [25] | 0 | 14.24 | 28.48 | 42.72 |
| Bhatia and Verma [26] | 5.696 | 14.24 | 11.392 | 31.328 |
| Li et al. [27] | 5.696 | 6.79 | 14.24 | 26.726 |
| Qu and Zeng [28] | 0 | 54.882 | 54.882 | 109.764 |
| Proposed Scheme | 2.848 | 17.088 | 11.392 | 31.328 |

### 6.2. Communication Cost

In Table 4, we compare the primary operations of communication cost for the proposed scheme and other schemes that were proposed, including those proposed by Yanfeng et al. [25], Bhatia and Verma [26], Li et al. [27], and Qu and Zeng [28]. As shown in Table 4, $NP_m$ represents the message size, which we assume was 2048 bits, $NP_q$ represents the parameter size, which belongs to elliptic curve and was equal to 160 bits [30,31], $NP_{ID}$

represents the identity size, which belongs to elliptic curve and was equal to 160 bits, and $NP_G$ represents the parameter size, which is part of the bilinear group, and its value was 1024 bits. As detailed in Table 4 and depicted in Figure 3, the proposed scheme had lower communication costs than its counterparts.

**Table 4.** Comparison of communication cost with major operations.

| Schemes | Signcryption Size | Signcryption Size in Bits |
|---|---|---|
| Yanfeng et al. [25] | $3\lvert NP_m\rvert + 4\lvert NP_{ID}\rvert + 12\lvert NP_q\rvert$ | 8704 |
| Bhatia and Verma [26] | $3\lvert NP_m\rvert + 4\lvert NP_{ID}\rvert + 11\lvert NP_q\rvert$ | 8544 |
| Li et al. [27] | $3\lvert NP_m\rvert + 6\lvert NP_{ID}\rvert + 10\lvert NP_q\rvert$ | 8704 |
| Qu and Zeng [28] | $3\lvert NP_m\rvert + 6\lvert NP_G\rvert$ | 12,288 |
| Proposed Scheme | $3\lvert NP_m\rvert + 5\lvert NP_q\rvert$ | 6944 |

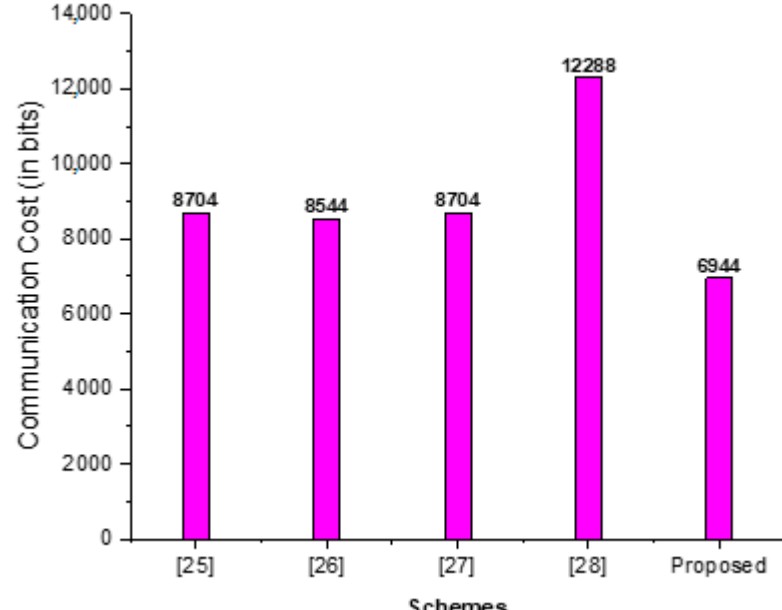

**Figure 3.** Comparison of the communication cost (in bits) of the proposed scheme and those proposed by Yanfeng et al. [25], Li et al. [27], and Qu and Zeng [28].

## 7. Conclusions

There is a growing trend toward integrating drones with B5G networks to meet the autonomy and pervasiveness requirements of future applications. Drones, however, have limited onboard storage and computational capabilities. Such restrictions make it difficult for a drone to execute cryptographic operations with a high level of complexity. Using the concept of elliptic curve cryptography (EEC) to resolve this shortcoming, we proposed a certificateless proxy signcryption scheme in this article. We performed a security analysis of the proposed scheme using the random oracle model (ROM) and demonstrated its resistance to well-known attacks. The proposed scheme had a significant advantage in that the partial private key can be disseminated over an open network without the risk of unauthorized disclosure. In terms of computational and communication costs, the proposed scheme's performance analysis was compared to existing schemes on the same topic. According to the findings of both studies, the proposed scheme outperformed its competitors in terms of security rigor and had a better security-to-efficiency tradeoff.

**Author Contributions:** Conceptualization, M.A.K., N.I. and I.U.; methodology., M.A.K., H.A., N.I. and S.A.H.M.; software, S.A.H.M., U.T. and W.A.; validation, M.A.K., H.A. and I.U.; formal analysis, I.U. and M.A.K.; investigation, H.A., N.I., I.U. and W.A.; resources, M.A.K., W.A. and S.A.H.M.;

data curation, W.A., N.I. and U.T.; writing—original draft preparation, M.A.K., I.U., W.A., N.I. and S.A.H.M.; writing—review and editing, M.A.K., H.A, U.T, W.A. and S.A.H.M.; visualization, U.T, W.A. and S.A.H.M.; supervision, M.A.K. All authors have read and agreed to the published version of the manuscript.

**Funding:** This research received no external funding.

**Data Availability Statement:** Not applicable.

**Conflicts of Interest:** The authors declare no conflict of interest.

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
