# Peer review of "A Resource-Friendly Certificateless Proxy Signcryption Scheme for Drones in Networks beyond 5G"

_drones, doi:10.3390/drones7050321_

Round 1

Reviewer 1 Report

In this article, the authors proposed a resource-friendly certificate-less proxy signcryption scheme for drones in networks beyond 5G, in which they allow the users and KGC to distribute the identity and partial private key in an open network without disclosing it to attackers. The article sounds good; however, the following minor suggestions must be addressed before publication.

• The authors need to justify how elliptic curve cryptography can achieve the goal of low cost. You mentioned proxy signcryption; however, there is less information about certificate-less proxy signcryption in the abstract. I suggest adding some motivational explanation about certificate-less proxy signcryption in the abstract.

• The authors need to explain the Random Oracle Model (ROM) so that readers can comprehend security analysis easily.

• The authors are required to include the adversarial or threat model.

•  The author included two adversary categories, Type 1 (???1) and Type 2 (???2), but gave a very short explanation that neglected to clarify their roles.

• The authors are suggested to add the diagram for the network model, which indicates the flow of the proposed scheme.

• Before the proposed scheme construction, the scheme syntax should be added.

• In the security analysis part, I found a few punctuation and grammar mistakes. The authors should carefully review it, who should then make the required changes.

Minor editing of English language required

Author Response

Response to Reviewer 1

We thank the reviewer for the time, effort, and valuable comments. We believe that all comments are constructive and contribute to enhancing the readability of our paper and enriching its contents. Therefore, all comments have been addressed appropriately. Please find the response and amendments made to your comments below.

Comment: In this article, the authors proposed a resource-friendly certificate-less proxy signcryption scheme for drones in networks beyond 5G, in which they allow the users and KGC to distribute the identity and partial private key in an open network without disclosing it to attackers. The article sounds good; however, the following minor suggestions must be addressed before publication.

Response: We appreciate the reviewer's recognition of our efforts and constructive feedback. We consider the suggestions of reviewers seriously and value their input.

Comment: The authors need to justify how elliptic curve cryptography can achieve the goal of low cost. You mentioned proxy signcryption; however, there is less information about certificate-less proxy signcryption in the abstract. I suggest adding some motivational explanation about certificate-less proxy signcryption in the abstract.

Response: HECC is an advanced version of the Elliptic Curve Cryptography. HECC is characterized by a smaller key size and, at the same time, promises security comparable to that of its counterparts, i.e., elliptic curve, bilinear pairing, and modular exponentiation. Incorporation of HEC reduces power consumption and improves the device’s performance, thereby making it suitable for a wide range of devices, ranging from sensors to drones. We have added a sentence regarding HECC in the abstract.

Comment: The authors need to explain the Random Oracle Model (ROM) so that readers can comprehend security analysis easily.

Response: We have explained the Random Oracle Model (ROM), please see the section 3.1.

3.1. Random Oracle Model

In 1993, Bellare and Rogaway created the Random Oracle Model (ROM). By considering hash functions as random oracles, this model makes it simple to verify the security of cryptographic algorithms that use hash functions.  In this paradigm, any input will result in an output of a predetermined length. If the input has been requested previously, the oracle returns the same value as it did previously. If the input is not one that the oracle has previously received, the oracle returns a randomly chosen output. You can substitute a hash function with an accessible random function (the "random oracle"). Therefore, an adversary must consult the random number generator to determine what the hash function will do.

Comment: The authors are required to include the adversarial or threat model.

Response: We have included the adversarial or threat model in the revised version, please see the section 3.2.

3.2. Adversarial or Threat Model

This section will outline potential security vulnerabilities that could compromise the confidentiality of the security parameter utilized in the generation of ciphertext and signatures. Two types of adversaries, namely Type 1 Type 1 () and Type 2 () are defined. The first type of attacker, denoted as Type 1 (), is an external threat actor who aims to compromise the confidentiality of the proposed scheme and engage in signature forgery. It should be noted that the entity denoted as () lacks the capability to access the private keys of the user, yet possesses the ability to replace the public key of said user.

The Type 2 () is the insider attacker (malicious) who desires to violate confidentiality and falsify the signature of the proposed scheme. It should be noted that the entity denoted as  possesses the capability to access the private key of ‘s, yet lacks the ability to substitute the public key of the user. The primary objective for both adversaries is to reveal the parameters utilized in the creation of the secret key and ciphertext. The subsequent objective entails the construction or retrieval of parameters utilized in the computation of a signature, followed by the generation of a forge signature.

Comment: The author included two adversary categories, Type 1 (???1) and Type 2 (???2), but gave a very short explanation that neglected to clarify their roles.

Response: We have explained both the adversaries and their roles in the revised version.

Comment: The authors are suggested to add the diagram for the network model, which indicates the flow of the proposed scheme.

Response: We have explained the network model in more detailed manners as shown in Figure 1. Changes have been highlighted.

Comment: Before the proposed scheme construction, the scheme syntax should be added.

Response: Before the proposed scheme construction, the scheme syntax has been added, please see the section 3.3.

Comment: In the security analysis part, I found a few punctuation and grammar mistakes. The authors should carefully review it, who should then make the required changes.

Response: As hinted, we have revised the write-up following an extensive grammatical check. In the security analysis part, we corrected the punctuation and grammar mistakes. Attempts have been made to assure spontaneity and lexical beautification.

Reviewer 2 Report

This information in this article is worthy of publication.  I do
feel the algorithm as presented advances the security items in
drone technology.  

There is a lot of small consistent English issues that I am not supposed
to correct but it needs an English review. I do feel my
comments need addressing before publication.

The biggest struggle, which is typical of algorithm discussion,
is how to present the algorithm in a readable fashion.  It is very
difficult for me to absorb all the abbreviations in the algorthm
description but I don't have an answer of how to do it better.
The algorithm discussion will lose reader interest because it is
so complex and difficult to get into presentable English.

B5G is not required for centimeter positioning.  Using a local GNSS base
or an RTN network usually in VRS mode the real-time centimeter
positioning has existed since approximately 2014.

GPS only is "old" technology.  Make reference to GNSS and explain
if other constellations have advantage of GPS in attacks.  In
addition the drone today would usually be outfitted with Inertial
Measuring Unit (IMU) which is less suspect to attacks and needs
to be mentioned.  Note IMU is covered later but should be added
to the GNSS dicussion in the introduction.

Line 170 Define whether Lidar is included when one says "cameras".

Section 3.2 It seems some of these processes exist in other
architectures thus some references are needed.

I do not see the need for the "hence proved" wording.

Line 295-296 I think something is missing after "private keys of".

I do not believe "Here, " is necessary in those sentences containing it.

Line 371 "massages" should be "messages" also line 439

Should Windows 7 be used as a testing standard as it is considered
"old" Windows technology?

Figure 2 I would leave [20] out as it forces all of the others
to not be as comparable graphically.

It is not discussed could current drone computational ability
handle this processing (I assume not) and what processing
capability will be required.  Does required hardware force the
drone to be heavier and more powerful battery dependent?

A lot of plural/not plural consistent issues and same English issues over and over again which will be caught by a competent English review.  Note I am not supposed to make English corrections

Author Response

Response to Reviewer 2

To begin, we would like to express our gratitude to the reviewer for their time, effort, and insightful remarks. We think that all of the comments are helpful and that they contribute to making our article easier to read and to improving the information that it contains. As a result, each and every remark has been responded to in an acceptable manner. The answer to your comments, as well as any adjustments that were made, may be seen below.

Comment: This information in this article is worthy of publication.  I do feel the algorithm as presented advances the security items in drone technology.  

Response: We are thrilled that the reviewer recognized our efforts and provided constructive feedback. We appreciate the suggestions of reviewers and are grateful for their valuable input.

Comment: There is a lot of small consistent English issues that I am not supposed to correct but it needs an English review. I do feel my comments need addressing before publication.

Response: Thank you for drawing our attention to this issue with your insightful review. As alluded to, we have performed a thorough grammar check. We will revise the final version of the manuscript to improve its English usage.

Comment: The biggest struggle, which is typical of algorithm discussion, is how to present the algorithm in a readable fashion.  It is very difficult for me to absorb all the abbreviations in the algorithm description but I don't have an answer of how to do it better. The algorithm discussion will lose reader interest because it is so complex and difficult to get into presentable English.

Response: In subsection 3.3, we have included a syntax of the proposed scheme, and it is our sincere hope that this will contribute to a deeper comprehension of the algorithm behind the proposed scheme. By carefully reading over the syntax at this point, one may have a deeper grasp of the scheme. As a result of the fact that the symbol table contains representations of all the symbols, the scheme will be easier to comprehend.

Comment: B5G is not required for centimeter positioning.  Using a local GNSS base or an RTN network usually in VRS mode the real-time centimeter positioning has existed since approximately 2014.

GPS only is "old" technology.  Make reference to GNSS and explain if other constellations have advantage of GPS in attacks.  In addition the drone today would usually be outfitted with Inertial
Measuring Unit (IMU) which is less suspect to attacks and needs to be mentioned.  Note IMU is covered later but should be added to the GNSS discussion in the introduction.

Response: We concur with the reviewer's assessment. This feedback motivated us to improve the quality of the article in the right direction. The reviewer's feedback was carefully considered and subsequently addressed. Please refer to the third paragraph of the introduction for relevant information, and notice that relevant references have been highlighted in the references section.

Comment: Line 170 Define whether Lidar is included when one says "cameras".
Response: We did not include Lidar, but we did discuss the cameras module. While cameras and Lidar are both sensors used for perception in robotics and autonomous vehicles, they operate on different principles and provide different types of information.
Comments: Section 3.2 It seems some of these processes exist in other architectures thus some references are needed.

Response: Relevant references are added.

Comment: I do not see the need for the "hence proved" wording.
Response: This phrasing has been deleted. I appreciate you bringing this to our attention.

Comment: Line 295-296 I think something is missing after "private keys of".

Response: The private keys of the user.  Correction has been made.

Comment: I do not believe "Here, " is necessary in those sentences containing it.

Response: This phrasing has been deleted. I appreciate you bringing this to our attention.

Comment: Line 371 "massages" should be "messages" also line 439

Response: Correction has been made.

Comment: Should Windows 7 be used as a testing standard as it is considered "old" Windows technology?

Response:  In the revised version, the computation cost has been evaluated utilizing the Raspberry board. Despite the availability of alternative replacements for RPI, which boast advanced hardware configurations such as LattePanda 4G/64 GB, Qualcomm Dragon board, ODROID-XU4, and ASUS Tinker Board, among others, RPI remains widely regarded as the most economical and power-efficient choice. Additional compelling attributes of the RPI 4 that reinforce its choice include its integrated wireless network capabilities, specifically, dual-band 802.11 b/g/n/ac Wi-Fi and Bluetooth 5.0 BLE. In the present scheme implementation, the model and hardware specifications are delineated as follows: The Raspberry PI 4B (2019) is equipped with a 64-bit CPU architecture and a 1.5 GHz quad-core processor. It operates on the Ubuntu 20.04.2 LTS operating system and has a memory capacity of 8 GB, as reported in reference [23]. represents elliptic curve scalar multiplications  and represents pairing operation. It has been observed with respect to average time that a single   takes 2.848 ms and 18.294 ms [23].

Comment: Figure 2 I would leave [20] out as it forces all of the others to not be as comparable graphically.

Response:  Since Figure 2 has been redrawn based on the newly calculated computation costs, it is now more visible.

Comment: It is not discussed could current drone computational ability handle this processing (I assume not) and what processing capability will be required.  Does required hardware force the drone to be heavier and more powerful battery dependent?
Response: Yes, currently drones (quadcopters) are typically equipped with Raspberry board. Despite the availability of alternative replacements for RPI, which boast advanced hardware configurations such as LattePanda 4G/64 GB, Qualcomm Dragon board, ODROID-XU4, and ASUS Tinker Board, among others, RPI remains widely regarded as the most economical and power-efficient choice.The Raspberry PI 4B (2019) is equipped with a 64-bit CPU architecture and a 1.5 GHz quad-core processor. It operates on the Ubuntu 20.04.2 LTS operating system and has a memory capacity of 8 GB.